# Limiting Network Bandwidth to Unleash Throughput for Serverless Systems

Prasoon Sinha*, Sidharth Babu†, and Neeraja J. Yadwadkar*

*University of Texas at Austin  *prasoon.sinha@utexas.edu, neeraja@austin.utexas.edu*
†Carnegie Mellon University  *sidhartb@andrew.cmu.edu*

*Abstract*—Serverless computing relieves developers from the burden of managing resources for their cloud applications. However, commercial providers require users to set a memory limit for their serverless function and then proportionally allocate (i.e., couple) the other resource types (CPU, network bandwidth).

A few works show the inefficiencies with coupling CPU and memory, and instead make independent allocations for the two resource types. However, despite right-sizing CPU and memory allocations, we empirically find that these systems fall short in meeting the desired throughput (function invocations per second or requests per second). We make a key observation that the throughput of serverless systems is limited due to network congestion. The root cause of this congestion is that existing systems ignore right-sizing network bandwidth for serverless functions, thereby increasing contention for this resource type.

In this work, we study commonly deployed serverless functions and find that network bandwidth is crucial to meet performance needs: a function's execution time can vary by $10\times$ depending on the amount of network bandwidth allocated. However, our analysis reveals that determining the required amount of network bandwidth to allocate is challenging: it depends on multiple factors, including the number of allocated CPU cores, function semantics, and function inputs.

To this end, we build SoloTune, a holistic resource management framework for serverless systems. SoloTune uses online learning to predict a function's compute time and then estimates the required network bandwidth to meet SLOs using an analytical model. Our initial experiments reveal that by just making intelligent network bandwidth allocations, we can reduce SLO violations by $1.3\times$ at high load compared to state-of-the-art solutions.

## I. Introduction

Serverless computing simplifies cloud usage for programmers: users simply upload their code while providers manage resources and auto-scaling on behalf of the users [1], [6], [16], [17], [19], [27]. However, commercial providers, including AWS Lambda [6] and Google Cloud Functions [16], require users to specify a memory limit for each function and allocate a proportional share of CPU and network bandwidth. Having to specify the memory limit forces users to reverse-engineer the implications of their specification on latency and cost.

Previous works attempt to automatically find the memory limit that will reduce user cost [9], [28]. A few works show the inefficiencies of proportional (i.e., "coupled") resource allocations and completely re-haul the resource allocation policy, making independent decisions for the amount of CPU and memory to allocate on behalf of the user [12], [41].

However, we empirically observe that despite right-sizing the amount of memory and CPU cores to allocate, these

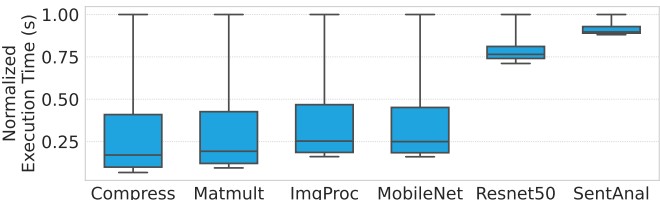

Fig. 1. Variation in execution time for six serverless functions as the amount of network bandwidth allocated varies from 16-1024 Mbps. The number of cores, amount of memory allocated, and function inputs are fixed. See § I.

systems still fail to meet the desired throughput (requests per second (RPS)) for function invocations. We make a key observation that *the root cause behind the limited throughput of serverless computing systems is often due to network congestion.*

Existing systems ignore right-sizing network bandwidth for each function, even when the most widely used functions heavily rely on this resource type to (1) download/upload data objects from external stores [13], [23], [37], and/or (2) transmit intermediate data between function workflows [38], [41].

Through our measurement study, we find that the amount of network bandwidth allocated can lead to a $10\times$ difference in execution time for the same function (Figure 1).

We study 11 representative serverless functions, including machine learning (ML) training/inference, scientific applications, image/video processing, and web services, from existing benchmark suites [13], [23], [37] widely used in previous works [4], [5], [15], [24], [25], [28], [29], [40].

We make two key findings: (a) not limiting network bandwidth usage quickly inflates execution time and increases service level objective (SLO) violations, as functions burst and consume large amounts of network bandwidth, creating contention for this resource type, and (b) coupling the amount of network bandwidth allocated with user-specific memory limits, as done by commercial providers [7], increases cost for the user and resource waste for the provider. Thus, it is imperative that *serverless systems right-size network bandwidth to improve the system's throughput while meeting SLO for functions.*

However, determining the amount of network bandwidth to allocate a function is challenging: our measurement study reveals that determining the network bandwidth and number of CPU cores needed for a serverless function is a joint optimization problem. The required amount of network bandwidth depends on the (a) number of cores allocated, (b) function

semantics, and (c) function inputs.

Using our insights, we introduce SoloTune, a resource management framework for serverless systems that makes independent allocation decisions for each resource type, considering function semantics and inputs, to meet latency SLOs. SoloTune leverages an online learning agent that uses linear regressors to predict a function's compute time (the time it spends executing on CPU resources) given an invocation's input. It then uses this prediction with an analytical model to estimate the network bandwidth required to meet SLOs.

We summarize our contributions as follows:

- We empirically show that not limiting the amount of network bandwidth a function can use or coupling the amount of network bandwidth allocated with memory is suboptimal: both lead to performance degradation and/or resource waste.
- We characterize real-world serverless functions and find that allocating network bandwidth depends on the number of cores allocated, function semantics, and inputs.
- We build SoloTune atop OpenWhisk using online learning with linear regression. Our preliminary results show that by simply making intelligent network bandwidth allocation per invocation, we can reduce SLO violations by $1.3\times$ at high load compared to state-of-the-art baselines.

## II. BACKGROUND & MOTIVATION

We describe the resource allocation policies of state-of-the-art serverless platforms. We then empirically show the inefficiencies of existing policies to motivate the need for better network bandwidth resource allocations.

### A. Existing Serverless Resource Management Policies

Commercial providers (e.g., AWS Lambda [6], Google Cloud Functions [16]) and open-source platforms (e.g., Open-Whisk [30], OpenFaaS [2]) use similar policies to allocate memory and CPU to serverless functions: they allocate a proportional CPU share to the user-specified memory limit.

However, these platforms differ in their policies for allocating network bandwidth. AWS Lambda continues to rely on resource coupling, allocating functions a network bandwidth limit proportional to the memory limit [7], while Open-Whisk/OpenFaaS do not place any limits, allowing functions to consume large amounts of network bandwidth.

State-of-the-art serverless resource management frameworks also ignore limiting network bandwidth.

Cypress [10], Golgi [24], and Kraken [11] provision containers with high CPU and memory capacity per function to consolidate multiple concurrent invocations in one container. OFC [29] leverages spare memory across functions to dynamically scale in-memory caches, reducing external data retrieval time.

While these systems couple CPU and memory, Bilal et al. [12] and Aquatope [41] decouple and make independent allocation decisions for CPU and memory. However, none of these systems limit network bandwidth, allowing functions to burst and consume large amounts of this resource type.

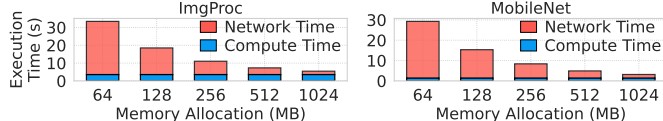

Fig. 2. Under a coupled resource allocation policy, allocating more memory reduces network time since the amount of network bandwidth allocated increases proportionately. However, this comes at the cost of memory underutilization. See § II-B.

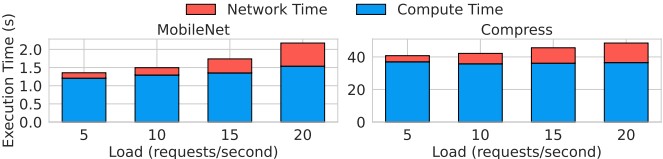

Fig. 3. Change in network and compute time as load increases for two functions placed on a single server without network bandwidth limits. Network time for both functions grows as network bandwidth bottlenecks. Compute time remains unchanged, as many cores remained unused. See § II-C.

Parrotfish [28] and AWS Lambda's Power Tuning [9] are offline profiling tools that select the best memory limit to minimize user cost. However, as these tools run atop AWS Lambda, they assume the platform's coupled allocation of network bandwidth, CPU, and memory. In the next sections, we show the inefficiencies of (1) coupling network bandwidth with memory, and (2) ignoring allocating network bandwidth altogether.

### B. Why Not Couple Network with Memory Allocations?

Previous works study the inefficiencies of coupling CPU and memory but ignore network bandwidth [12], [41]. We extend these studies and evaluate the impact of coupling network bandwidth with memory. Figure 2 shows the execution time for two functions (ImgProc, MobileNet) as we vary the amount of memory allocated. We use AWS' policy [7] to allocate network bandwidth in proportion to the amount of memory allocated. We break down execution time into (1) network time, the time the function downloads/uploads data from/to external datastores like S3, and (2) compute time, the time the function executes on CPU cores.

ImgProc takes $\sim 30$ seconds to download its 30MB image when allocated 64MB memory, as it is allocated 8 Mbps network bandwidth. This download time alone violates the SLO (7 seconds). If the request's compute time is 4.4 seconds, download time must complete within 2.6 seconds to meet the SLO. Hence, the request requires 92 Mbps (30MB × 8 bits / 2.6 sec). To obtain an allocation of 92 Mbps network bandwidth, users need to request > 512MB of memory under a coupled allocation policy. However, ImgProc only consumes at most 59MB, resulting in > 88% memory underutilization. **Takeaway #1:** Serverless systems need to make independent allocation decisions for network bandwidth and memory to reduce resource waste for the provider while meeting performance requirements for the user.

### C. Why Limit Network Bandwidth Usage?

Although we show the importance of independently allocating network bandwidth and memory, we next evaluate whether

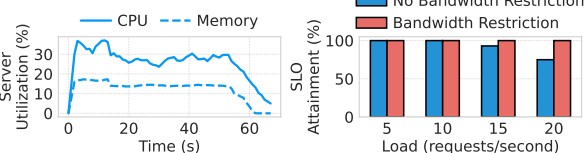

Fig. 4. Left: Server CPU/memory utilization at RPS 20. These resources are not the cause of SLO violations (<50% util). Right: Placing network bandwidth limits per function improves SLO attainment. See § II-C.

restricting network bandwidth is necessary; we could simply let the host OS (e.g., Linux) multiplex network bandwidth across concurrent requests without placing any restrictions. To do this, we observe the SLO attainment of functions on OpenWhisk with and without network bandwidth limits as we increase load (RPS). We deploy OpenWhisk on a cluster of four bare-metal servers, each containing 96 cores and a 10 Gbps NIC. We use Azure's production serverless trace [33] to create a realistic invocation arrival pattern (see § V-A workload creation). We set each function's SLO to $1.5\times$ its execution time observed in isolation and execute only single-threaded functions with a 1-core allocation to ensure suboptimal core allocations are not the cause of SLO violations.

Figure 3 shows the average network time and compute time as load increases for two functions, MobileNet and Compress, that OpenWhisk places on the same server. Both functions' compute time only slightly increases with load, suggesting that CPU resources are not saturated even at RPS 20. Figure 4a confirms this: server CPU and memory utilization is <40% at RPS 20. However, network time grows by 3-4$\times$, indicating the server's 10 Gbps NIC bottlenecks. This manifests into several SLO violations: when not restricting bandwidth, SLO violations begin when RPS >10 and SLO attainment drops by 25% at merely RPS 20 (Figure 4b). While all the requests to Compress meet SLOs despite its inflated network time, every request to MobileNet violates SLOs at RPS 20. To meet its SLOs, MobileNet requires >448 Mbps to ensure network time <0.65 seconds (2-second SLO − 1.35-second compute time). However, MobileNet is only given 375 Mbps on average, since the requests to Compress, which download larger input files, use 600 Mbps on average and saturate the NIC, despite only requiring 460 Mbps to meet SLOs. By placing the optimal network bandwidth limits on each function (MobileNet 448 Mbps, Compress 460 Mbps), we do not violate any SLOs, even at RPS 20, thereby doubling the effective throughput the serverless system can support (Figure 4b).

**Takeaway #2:** To meet the throughput demands of serverless workloads while also meeting SLOs across users and functions, serverless systems should allocate only the required amount of network bandwidth to each function invocation.

## III. CHARACTERIZATION

In § II, we motivate the need to intelligently allocate network bandwidth. While we fixed the amount of cores allocated in § II (1 core), we next study the impact of different combinations of network bandwidth and CPU allocations on execution time. We also study the impact of function inputs on network and compute time. We omit memory from our analysis; memory does not affect latency since, traditionally, serverless platforms do not provide swap space [6], [16], [27].

**Impact of network bandwidth & CPU allocations.** Figure 5 presents the execution time of six representative functions under varying amounts of network bandwidth and CPU cores allocated. We keep each function's input fixed to isolate the impact of variable allocations on execution time.

The effect of both resource types on Matmult and VidProc's execution time shows that for multi-threaded functions, determining the amount of network bandwidth and cores to allocate is a joint optimization problem. For example, to achieve a 150-second execution time, we could allocate Matmult 3 cores with 512 Mbps, 4 cores with 384 Mbps, or 8 cores with 256 Mbps. Increasing the amount of network bandwidth and cores allocated significantly reduces execution time: Matmult's latency reduces by 25% when increasing the cores allocated from 2 to 16 and further reduces by 75% when increasing the network bandwidth allocated from 64 to 512 Mbps.

However, for other multi-threaded functions (e.g, MLTrain, ResNet50), increasing the amount of network bandwidth allocated has little impact on execution time: given 16 cores, increasing network bandwidth from 64 to 512 Mbps only reduces latency by 15% for MLTrain and 11% for ResNet50. Meanwhile, these functions greatly benefit from more cores: MLTrain's execution time reduces by 58% when increasing cores allocated from 2 to 16 with a fixed 64 Mbps network bandwidth. These types of functions may not require larger amounts of network bandwidth to meet SLOs.

Unlike the multi-threaded functions, increasing the number of cores allocated for ImgProc and Compress has little effect on execution time (<0.01% variation) as these functions are single-threaded. Meanwhile, providing more network bandwidth greatly impacts execution time: ImgProc's latency reduces by 49% by increasing network bandwidth from 64 to 512 Mbps.

Meeting SLOs for single-threaded functions likely requires generous network bandwidth allocations—insufficient allocations due to contention or poor provisioning can rapidly lead to violations.

**Takeaway #3:** The impact of allocated network bandwidth and CPU cores is function-specific. The required network bandwidth heavily depends on the allocated number of CPU cores and function semantics (thread-bounded parallelism).

**Impact of function inputs.** We next observe the impact of varying inputs on network time and compute time. We fix the amount of network bandwidth (256 Mbps) and CPU (1 core) allocated across inputs. Figure 6 shows this data for VidProc; however, our findings hold for other functions. Across all functions, the only input property affecting network time is an input's size: downloading two videos of the same size (2.2 MB) takes the same amount of time (0.07 seconds). However, their compute times differ greatly: two videos of the same size differ by up to 3.02$\times$ in compute time. Videos in Set 1 exhibit a seemingly unpredictable relationship between input size and execution time, while Set 2 shows a steady growth in compute

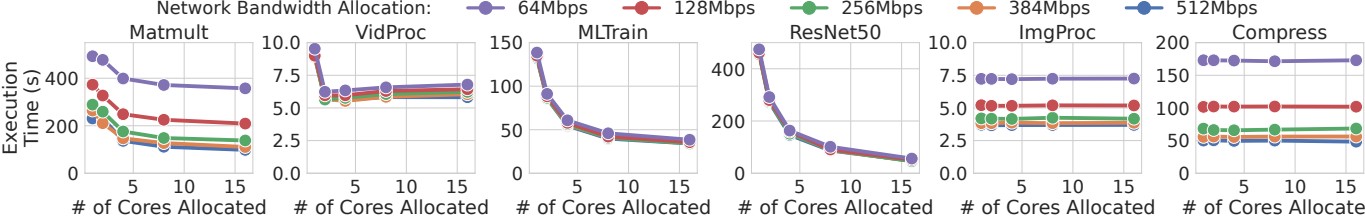

Fig. 5. Change in execution time of six serverless functions as the amount of network bandwidth and number of cores allocated vary. See § III.

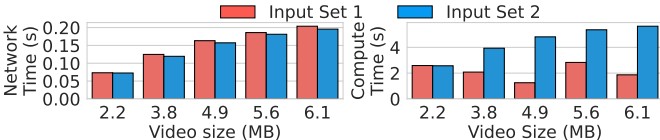

Fig. 6. Network time and compute time of two sets of inputs to VidProc. Both sets contain the same size, but different unique inputs. Each input is given 256 Mbps and 1 core. See § III.

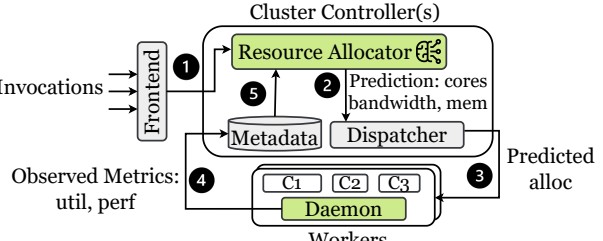

Fig. 7. SoloTune's architecture (green components) and workflow. See § IV.

time with video size. We found that all videos in Set 2 have the same resolution (1280x720), whereas the resolution varies in Set 1 videos, causing wide variation in execution time.

**Takeaway #4:** Serverless resource managers need to be input-aware. While the amount of required network bandwidth depends on just input size, determining the number of CPU cores needed requires being holistically input-aware (i.e., we need to consider characteristics of inputs beyond just size).

## IV. DESIGN & IMPLEMENTATION

We leverage our findings from § II and III to build SoloTune, a resource management framework for serverless systems that makes input-aware and independent allocation decisions for all three resource types: network bandwidth, CPU, and memory. SoloTune provides the minimum resource requirements to meet each function's latency SLO.

**SoloTune's workflow.** Figure 7 describes SoloTune's workflow. ❶ The Frontend receives invocations specifying the function, input(s), and a latency SLO. It then forwards these invocations to SoloTune's Resource Allocator. ❷ To make input-aware allocation decisions, the Allocator extracts readily available metadata (e.g., image resolution) specific to the input type in the payload and feeds this metadata as features to its function-specific online machine learning (ML) agent to predict the required resource allocation to meet the latency SLO. ❸ The Dispatcher then places the invocation with the predicted resource allocation on a server with available resources. ❹ On each worker, SoloTune deploys a lightweight daemon to collect performance metrics per invocation. When an invocation completes, the daemon sends this data to

❺ SoloTune's Allocator to update its model and learn the evolving relationship between inputs, allocated resources, and function performance.

**Allocating network bandwidth.** To determine the required network bandwidth for an invocation, we break down the E2E latency of serverless invocations. Traditional serverless functions have two distinct phases of execution: time using network bandwidth to access the inputs, and time using CPU cores. Hence, the E2E latency of a serverless invocation can be formally decomposed as

$$L_{e2e}(i) = L_n(i) + L_c(i) \qquad (1)$$

where $L_{e2e}(i)$ is the E2E latency for the given input $i$, $L_n(i)$ is the network time, and $L_c(i)$ is the compute time. For an invocation to meet its SLO, SoloTune needs to allocate enough resources to ensure $L_{e2e}(i) \leq \alpha \times SLO(i)$, where $0 < \alpha \leq 1$ to provide some slack. Substituting this into Equation (1) and rearranging terms, we solve for network time $L_n(i)$ as

$$L_n(i) = \alpha \times SLO(i) - L_c(i) \quad \text{where } 0 < \alpha \leq 1 \qquad (2)$$

We also know the network time equals the input's size $s_i$ divided by the amount of network bandwidth allocated $BW_{alloc}(i)$ (Figure 6a). Hence, we can determine the required amount of network bandwidth to meet the SLO for input $i$ given it has a size $s_i$ MB and $L_c(i)$ compute time as

$$BW_{alloc}(i) = s_i \times (\alpha \times SLO(i) - L_c(i))^{-1} \qquad (3)$$

We can retrieve the input's size upon invocation time by analyzing its properties; however, estimating the compute time before runtime requires understanding the effect of inputs *and* cores allocated on compute time (§ IV-A).

**Allocating CPU and memory.** In this work, we focus on predicting compute time for commonly used single-threaded functions [4], [5], [13], [15], [23]–[25], [28], [29], [40] that utilize only one core.

Hence, the number of cores allocated is fixed; we only model the effect of different inputs on compute time. We leave predicting compute time for multi-threaded functions allocated multiple cores as future work. We also leave predicting memory as future work. We describe how we predict compute time for single-threaded functions next.

### A. Predicting Compute Time

We approach predicting compute time as a regression problem. We use online learning that leverages linear regression to predict compute time.

**Why online ML?** § III highlights the complexity of making accurate resource allocations. Input properties beyond size (e.g., video resolution) affect compute time. Moreover, the impact of the number of cores allocated on performance can greatly differ between functions due to differences in function semantics. Thus, we use ML to predict the compute time per invocation in a data-driven manner. We further design SoloTune's Resource Allocator to use *online* ML for four reasons. (1) Online ML enables our agent to observe and adapt to dynamically changing runtime environments. (2) Representative function inputs may not be available offline to train accurate models. (3) Models trained offline are susceptible to data drift if the distribution of inputs, functions, or SLOs changes over time. (4) Training ML models offline may not generalize to new, previously unseen functions and inputs.

**Inputs to online agent.** The input to SoloTune's online agent is an invocation's input. We construct input-specific feature vectors consisting of readily available metadata commonly used to describe input types (e.g., video/image resolution, file size, bit rate). Using readily available metadata ensures fast and resource-efficient vector construction. Table I lists the input types SoloTune extracts metadata for; these input types are commonly seen in commercial serverless applications [8] and research [4], [5], [10], [12], [23], [24], [28], [29], [37], [39]. This metadata helps the agent learn input features affecting a function's execution time. SoloTune infers the input type from the file extension if not specified. If SoloTune has not seen the input type (e.g., XML), developers can specify the input's descriptive features during function registration; otherwise, SoloTune defaults to input size as the feature. SoloTune concatenates the input's metadata to construct its feature vector for model prediction. We assume SoloTune can access input metadata since inputs are embedded in an invocation's payload. However, if not embedded, we can deploy SoloTune's featurizer as a small service that users submit inputs to for featurization.

**Feedback and model updates.** We update SoloTune's agent after each retired invocation using the observed compute time (execution time - input size / bandwidth allocation). SoloTune bootstraps its online agent by observing the first few invocations under intentionally large allocations: it does not use its compute time predictions to make network bandwidth allocation decisions. These initial invocations help the agent learn the relationship between inputs and execution time without negatively affecting user invocations. A confidence threshold determines how many invocations to observe before switching to predictive allocations.

### B. Implementation

We implement SoloTune on Apache OpenWhisk (OW) [30]. SoloTune's Resource Allocator is a shim layer atop OW, sitting on the same node as the dispatcher. SoloTune's online agent uses Vowpal Wabbit's lightweight and efficient linear regressors [3]. We write simple C++ functions to obtain input metadata within $\mu$s. For initial invocations (20 invocations), we default the amount of network bandwidth allocated (256

| Functions | # Threads | # Inputs | Input Sizes |
|---|---|---|---|
| Encrypt | single | file (36) | 2-24KB |
| SentAnal | single | json (23) | 63KB-5MB |
| ImgProc | single | image (64) | 184KB-4.5MB |
| ResNet50 | multiple | image (64) | 184KB-4.5MB |
| MobileNet | single | image (64) | 184KB-4.5MB |
| VidProc | multiple | video (28) | 2.2-6.1MB |
| Linpack | multiple | matrix (22) | 5.3MB-1.7GB |
| Speech2Txt | multiple | audio (32) | 48KB-12MB |
| Compress | single | file (28) | 64MB-2GB |
| Matmult | multiple | matrix (25) | 5.3MB-1.7GB |
| MLTrain | multiple | csv (4) | 10-100MB |

TABLE I
FUNCTIONS USED IN THIS WORK. SEE § V-A.

Mbps) while the agent learns. We limit the network bandwidth of containers using Linux's token bucket filter queuing discipline [36] and augment OW to support allocating network bandwidth per invocation. SoloTune's per-worker daemon collects execution time per invocation and sends it (via gRPC) to SoloTune's Allocator to update its agent online.

## V. EVALUATION

In this section, we provide preliminary results that show the efficacy of making intelligent network bandwidth allocations to better meet the throughput demands of serverless workloads while reducing SLO violations across loads and functions.

### A. Evaluation Methodology

**Testbed.** We deploy SoloTune on 4 servers in Chameleon Cloud [22] connected via a BCM57414 NetXtreme-E 10Gb RDMA Ethernet Controller. Each server has two Intel Xeon Gold 6240R CPUs (96 cores) at 2.40 GHz, 192GB memory, and runs Ubuntu LTS 20.0.4. One server hosts OpenWhisk's Controller, CouchDB, Dispatcher, and SoloTune's central components (allocator and metadata store). The other servers host an OpenWhisk Invoker and SoloTune's daemon.

**Functions.** Serverless production traces [20], [32], [33] conceal the details about actual functions executed. Thus, we use representative functions (Table I) from several serverless benchmark suites [13], [23], [37] commonly used in recent works [4], [5], [10], [12], [15], [18], [24], [25], [28], [29], [35], [40]. As we focus on single-threaded functions in this work, we use five functions: MobileNet, ImgProc, Compress, SentAnal, and Encrypt. We use inputs with multiple descriptive features (e.g., size, resolution).

**Workload.** We follow the methodology of previous works [10], [21], [24], [34], [35] to create our workload. We use Azure's production serverless trace [33] to evaluate SoloTune with realistic arrival patterns. As the trace only provides timestamps of arriving invocations without detailing the function/input, we select a function/input uniformly at random for each timestamp. We set every function's SLO to $1.5\times$ its median observed execution time. Like commercial providers [14], [26], [31], we generate low, medium, and high load to reflect cluster utilization of 25% (20 RPS), 50% (60 RPS), and 75% (100 RPS).

**Baselines.** We compare SoloTune against two state-of-the-art baselines: Parrotfish [28] and Bilal et al. [12]. As Parrotfish runs atop AWS Lambda, it couples network bandwidth with its recommended memory allocation per function. Meanwhile,

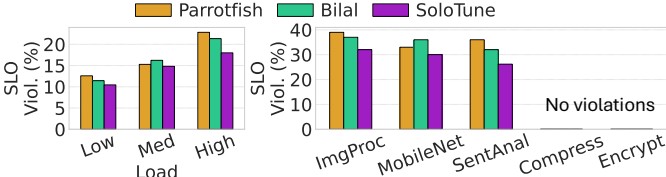

Fig. 8. By only making intelligent network bandwidth allocations (core allocations are the same across systems, 1-core), SoloTune reduces the SLO violations across loads (left plot) and functions (right) by 1.3×. See § V-B.

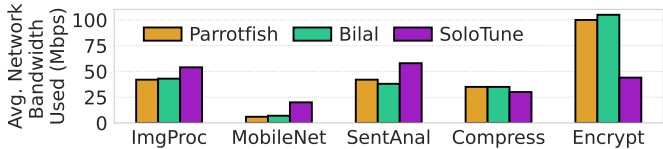

Fig. 9. SoloTune redistributes network bandwidth across functions to better meet SLOs (Figure 8). SoloTune limits the bandwidth usage of Encrypt and Compress to allocate more bandwidth to the other three functions. See § V-B.

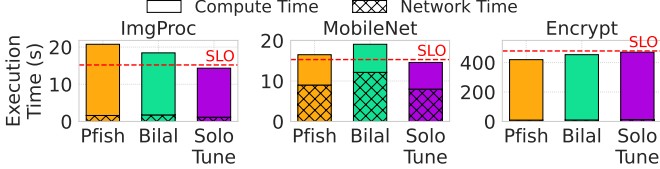

Fig. 10. SoloTune reduces ImgProc and MobileNet's network time by 1.6× and 1.5×, respectively, while slightly increasing Encrypt's by 1.3× to meet SLOs across all functions. Other functions omitted for brevity. See V-B.

Bilal et al. make independent allocation decisions for CPU and memory per function, but do not restrict bandwidth.

### B. Evaluation Results

**Preliminary E2E results.** By simply making intelligent network bandwidth allocations, SoloTune reduces SLO violations across loads (1.3× reduction in violations at high load, Figure 8, left). Moreover, SoloTune's allocations improve SLO attainment across all functions (Figure 8, right). None of the systems violate Compress or Encrypt's SLOs: these are longer-running functions with more slack. However, Parrotfish and Bilal increase the SLO violations for the other three functions by 26% compared to SoloTune.

Under Parrotfish and Bilal, Encrypt is able to consume > 100 Mbps of bandwidth on average (Figure 9). Hence, the plethora of requests to this function (nearly) saturate the server's bandwidth capacity. This reduces the available bandwidth for ImgProc, MobileNet, and SentAnal, increasing network time and hence overall latency/SLO violations for these functions. Meanwhile, SoloTune reduces the amount of network bandwidth allocated to Encrypt by over > 2× compared to the baselines; it learns that the function can meet SLOs even with small allocations (48 Mbps, Figure 10), as the slack between the function's SLO and compute time is high. Moreover, to meet their SLOs, SoloTune increases the allocation for ImgProc, MobileNet, and SentAnal (1.2-2× increase compared to the baselines) to decrease network time (Figure 10), as these functions have tighter latency constraints.

**Accuracy.** SoloTune's online agent uses regression to predict an invocation's compute time for a given function input.

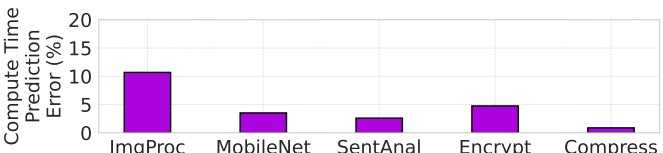

Fig. 11. Prediction error of SoloTune's online agent predicting compute time per invocation. Error is reported as mean absolute percentage error. See § V-B.

Figure 11 shows the agent's prediction error reported as the mean absolute percentage error for the invocations sent to each function. SoloTune's predictions are accurate (<11% error).

Serverless functions typically complete a single task without complex control logic that cause large variations in execution flow for a given input. This enables SoloTune to quickly and accurately learn the impact of different inputs on the function's compute time.

**Overheads.** SoloTune's input featurization and model prediction are on an invocation's critical path. However, we minimize these overheads. Regardless of the input type, featurization only takes 20-400$\mu s$. Model predictions are always $< 16\mu s$, and model updates take 100-300$\mu s$, however, updates are not on an invocation's critical path. The resource overheads are also minimal. The input feature vectors are 40-70 bytes. The weights comprising each function's regressor are at most 20 bytes (<one-millionth a server's memory). The CPU time for input featurization and model prediction/updates is $< 400\mu s$.

### VI. CONCLUSION, LIMITATIONS, AND FUTURE WORK

To improve the throughput and SLO attainment of serverless systems, we show the importance of making intelligent network bandwidth allocations. We present SoloTune, a resource management framework for serverless systems that uses online learning to predict a function's compute time and then estimates the required network bandwidth to meet SLOs using an analytical model. Our initial prototype reduces SLO violations by 1.3× compared to state-of-the-art solutions, simply by making intelligent network bandwidth allocations. However, we make a few assumptions that require further research.

**Future work.** (1) Our network bandwidth allocation decisions assume unidirectional usage of a server's network to download data objects before compute time. However, functions may also transmit data to a datastore or subsequent functions after compute time, which adds to the function's execution time. Accounting for this when making network bandwidth allocation decisions requires knowledge of the data size transmitted, which is unknown before runtime. We will conduct further research to predict the transmitted data size before runtime. (2) In this work, we focus on providing the required network bandwidth for single-threaded functions by predicting their compute time when only one core is allocated to them. For multi-threaded functions, we show that determining the network bandwidth and number of cores to allocate is a joint optimization problem (§ III). Moreover, predicting the compute time for multi-threaded functions is challenging, as the impact of multi-core allocations is function- and input-specific. We will build new techniques to predict compute time for multi-threaded functions and navigate the joint optimization

allocation problem. (3) We show SoloTune's initial gains while using OpenWhisk's memory-centric scheduler, which packs invocations onto a single server until memory is saturated. Hence, the scheduler continues packing invocations onto a single server and bottlenecks CPU/network bandwidth, thereby increasing SLO violations.

Further work is required to build a hierarchical, resource-aware scheduler that intelligently disperses invocations across servers to ensure no single resource type bottlenecks and causes unnecessary SLO violations. This is especially challenging for serverless computing, as functions exhibit vast differences in execution time and utilization of different resource types (§ III).

## Acknowledgements

We thank the anonymous reviewers for their helpful feedback. We thank the members of the UT-SysML research group for their insightful discussions to improve this work. This work was supported by the UT ECE junior faculty start-up fund, UT iMAGiNE consortium and its industrial affiliates, an award from the UT Machine Learning Lab (MLL), the AMD Chair Endowment, the Cisco Research Award, and the Amazon Research Award.

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
