# OpenReview forum: "Limiting Network Bandwidth to Unleash Throughput for Serverless Systems"
_iscaconf.org/ISCA/2025/Workshop/MLArchSys — MLArchSys 2025 Oral_

### Official Review · Reviewer_x1yB · 2025-05-13
**An Interesting Resource Allocation Problem with Limited Modeling Accuracy and Practical Contribution**

**Confidence:** 4
**Rating:** 5

**Detailed Feedback And Questions For Authors:**

The paper tackles an important problem with a sound approach and encouraging initial findings. However, its practical impact is substantially limited due to modest novelty and critical shortcomings, such as the restriction to single-threaded functions and an overly simplified model of network and computation.

1. "Proportional allocation" and "coupling": The abstract and introduction should more clearly define what the authors mean by "proportionally allocate (i.e., couple) the other resource types." Specifically, the paper needs to explicitly justify why "proportional allocation" is considered equivalent to "coupling" in this context.

2. Figure 1: The authors should clarify the presentation in Figure 1. Does it illustrate, for each function, the variation in normalized execution time across the entire specified bandwidth range (16-1024Mbps), or does it represent performance at a single, representative value within that range?

3. "joint optimization problem": The paper states that determining "the network bandwidth and number of CPU cores needed... is a joint optimization problem." To strengthen this point, the authors should clarify what non-joint or simpler approach this is being contrasted with, as the statement on its own might appear self-evident to some readers.

4. Introductory insights: The insights and observations presented in the introduction appear to have some overlap. Consolidating these points could improve the section's conciseness and impact. In addition, there are repetitive elements across Sections I and II. These sections could be more concise. Furthermore, a clearer and more explicit definition of how "coupling" applies to memory and bandwidth should be provided early on, ideally in Section I, to ensure reader understanding.

5. Takeaways #3 and #4: Observations such as those in Takeaway #3 (network/CPU impact is function-specific) and Takeaway #4 (input-awareness for network vs. CPU) might be perceived by some as intuitive or not significantly novel. The authors could consider merging these takeaways and focusing more prominently on the more innovative aspects and findings of their work.

6. A general comment on the methodology: The current approach often isolates individual resource components (e.g., memory, bandwidth) for analysis. A more comprehensive insight into resource interplay might be gained by allowing all relevant parameters to vary simultaneously. This would enable the formulation of an optimization problem (potentially solvable with black-box optimizers) to identify the optimal set of parameters for minimizing run time. Analyzing parameters one at a time can provide a limited perspective on their complex, combined effects.

**Top Reasons To Accept The Paper:**

1. Problem Identification: The paper compellingly argues that network bandwidth is a first-class resource bottleneck in serverless systems that has been largely ignored or mishandled by previous work focusing on CPU and memory. The empirical evidence showing a 10x variation in execution time based on network bandwidth and the demonstration that network congestion is a primary cause of SLO violations strongly motivates the need for the proposed research. This addresses a timely and practical issue in serverless platform design.

2. Promising Preliminary Results: SoloTune's approach of using online learning to predict compute time and then applying an analytical model to derive network bandwidth requirements is a practical and relatively lightweight solution. The preliminary results are promising, showing a 1.3x reduction in SLO violations at high load.

**Top Reasons To Reject The Paper:**

1. Oversimplified Model: The analytical model for network bandwidth allocation (Equation 3) assumes a simple "download data, then compute" execution pattern. It does not account for functions that might have interleaved network I/O and computation (which is very common e.g. for AI workloads), or functions that need to upload significant amounts of data after computation (e.g., image processing results, ML model updates). This is acknowledged (to some extent) as a limitation (Future Work #1) but in practice this means that the approach suggested by the authors is highly unlikely to contribute to real-world use cases.

2. Limited Scope: The current implementation and evaluation of SoloTune focus exclusively on single-threaded functions with a fixed 1-core allocation. While the characterization study highlights the importance and complexity of jointly optimizing CPU cores and network bandwidth for multi-threaded functions, the proposed solution defers this crucial aspect to future work. This significantly limits the generalizability and practical impact of the current system, as many real-world serverless functions are or could be multi-threaded.

---

### Official Review · Reviewer_2Nc1 · 2025-05-18
**No new insights are offered and the authors do not acknowledge basic literature on the topic of resource management..**

**Confidence:** 4
**Rating:** 3

**Detailed Feedback And Questions For Authors:**

* This paper offers no new insights; conclusions are obvious
* Benchmark set is very narrow
* The paper could easily be shortened by a factor of 2 without any loss of information. The write-up is repetitive.
* No reference is made to basic literature on resource management.

**Top Reasons To Accept The Paper:**

None

**Top Reasons To Reject The Paper:**

This paper addresses the classical problem of resource management in a serverless context. No new insights are offered and the authors do not acknowledge basic literature on this topic.

---

### Official Review · Reviewer_qHJt · 2025-05-19
**The paper addresses the problem that not accounting for network bandwidth in serverless function provisioning leads to sub-optimal performance, even when CPU and memory are right-sized. To address this, it proposes a joint optimization that allocates network bandwidth alongside CPU and memory, using online learning to predict compute time and analytically determine bandwidth.**

**Confidence:** 4
**Rating:** 6

**Detailed Feedback And Questions For Authors:**

Thank you for submitting the paper to MLArchSys. I enjoyed reading the clearly written paper.

- While the paper acknowledges single-threaded evalutation, it would be helpful to clarify whether the container and network stack share the same core, and how this affects the reported end-to-end latency. For example, using a single core for both function execution and network processing could introduce additional contention and latency, especially as request rates increase.
- When Fig. 2 shows increasing memory allocation, from the text it appears that this also increases the CPU allocation for the container. If so, by how much?

- The paper acknowledges that for some multi-threaded workloads, increasing bandwidth has little effect compared to increasing core count. What are the workloads that are evaluated in Fig. 8 to Fig. 10? The methodology section states "We follow the methodology of previous works [10], [21], [24], [34], [35] to create our workload".Clarifying this in the methodology is necessary, as right now it is not explicitly stated.

- The paper mentions using a confidence threshold for the online model but quantifying the effects to cold-start and on SLOs would be valuable, especially given the importance of tail latency in serverless.

- The paper can improve by differentiate its approach from recent works on network resource isolation, networking stack design, and network latency in serverless. A few interesting prior works that address the problems in network resource isolation, networking stack design, and network latency in serverless:
-- Network Resource Isolation in Serverless Cloud Function Service
-- I/O Resource Isolation of Public Cloud Serverless Function Runtimes for Data-Intensive Applications
-- In-Storage Domain-Specific Acceleration for Serverless Computing
-- Aquatope: QoS-and-Uncertainty-Aware Resource Management for Multi-Stage Serverless Workflows
-- Rethinking the Networking Stack for Serverless Environments
-- FaaSTube: Optimizing GPU-oriented Data Transfer for Serverless Computing
-- FasDL: An Efficient Serverless-Based Training Architecture With Communication Optimization and Resource Configuration

**Top Reasons To Accept The Paper:**

- The paper convincingly demonstrates, both analytically and empirically using prior production traces, that network bandwidth is a critical bottleneck in serverless systems. The characterization is comprehensive.
- The paper also lists the limitations which scopes the work.
-  Use of online learning for per-invocation compute time prediction enables the system to adapt to diverse input characteristics and function semantics.

**Top Reasons To Reject The Paper:**

- The setup that uses only single-threaded experiments and methodological gaps, such as lack of detail on workload composition, unclear CPU/memory scaling in experiments. These limit the generality of the contribution.

---

### Official Review · Reviewer_Fi2E · 2025-05-19
**Review of Submission2**

**Confidence:** 4
**Rating:** 5

**Detailed Feedback And Questions For Authors:**

## Summary

The paper solves a critical problem of intelligent resource allocation for serverless computing services. It emphasizes the need to decouple resource allocation, particularly network bandwidth from other resources such as compute and memory. The authors describe various scenarios where coupling or even unrestricted resource allocation can negatively affect the Service level Objectives(SLOs) for certain workloads. Furthermore, the work highlights the importance of function specific allocation and input characteristics for serverless resource managers.

To this end, the work suggests a simple, low-overhead method of using an ML agent that predicts the resource allocation based on the function input metadata


## Comments

Overall I like the theme of this work in effectively underscoring the importance of decoupling or constraining the network bandwidth resource to ensure optimal usage across various functions. I agree with all the takeaways proposed in Sections II and III. I have the following comments around  the ML agent.

1. In Section IV, Solotune's workflow states that the ML agent uses the metadata of the payload to predict resource allocation. Figure 7 shows predictions as cores, bandwidth and memory. However, Section IV.A."Feedback and model updates" only refer to the compute time prediction by the ML agent. From my understanding, it appears the prediction is limited to the compute time, which is later used to estimate bandwidth allocation analytically. It would be helpful to explicitly state this.

2. I am also not satisfied with the prediction accuracy. The current approach lacks consideration of memory allocation, and the worker state. Suppose, a worker is already memory-constrained and the function is memory bound, the latency of the function could be incorrectly predicted. Incorporating these factors could enhance prediction accuracy.

3. What is the rationale for not adopting a dedicated prediction agent for each function? I would expect that function-specific prediction agents would greatly improve the accuracy of compute time prediction.

**Top Reasons To Accept The Paper:**

The paper suggests a resource management framework called SoloTune, that takes in the function and its input/metadata to predict compute times using an online ML agent. The compute time estimate is later used to estimate required bandwidth to satisfy the SLO for the corresponding function. SoloTune also takes in the performance metrics for previous invocations to update the ML agent. The operational overheads for the SoloTune approach are tiny.

This works presents a viable solution for prevalent serverless systems, showing significant potential for enhancing the quality of service compared to current approaches.

**Top Reasons To Reject The Paper:**

A more thorough explanation of the ML agent, including a detailed breakdown of its behavior, is needed to justify its use over a purely analytical method. The current workload only demonstrates improvements for single-threaded functions.

Additionally, sensitivity studies on the alpha parameter in Equation (2) and its effect on tail latency would be useful.